# Retinal Pigment Epithelium Pigment Granules: Norms, Age Relations and Pathology

**DOI:** 10.3390/ijms25073609

**Published:** 2024-03-23

**Authors:** Alexander Dontsov, Mikhail Ostrovsky

**Affiliations:** Emanuel Institute of Biochemical Physics, Russian Academy of Sciences, Moscow 119334, Russia; adontsovnick@yahoo.com

**Keywords:** retinal pigment epithelium, reactive oxygen species, lipofuscin, bisretinoids, oxidative stress, melanin, melanolipofuscin, aging

## Abstract

The retinal pigment epithelium (RPE), which ensures the normal functioning of the neural retina, is a pigmented single-cell layer that separates the retina from the Bruch’s membrane and the choroid. There are three main types of pigment granules in the RPE cells of the human eye: lipofuscin granules (LG) containing the fluorescent “age pigment” lipofuscin, melanoprotein granules (melanosomes, melanolysosomes) containing the screening pigment melanin and complex melanolipofuscin granules (MLG) containing both types of pigments simultaneously—melanin and lipofuscin. This review examines the functional role of pigment granules in the aging process and in the development of oxidative stress and associated pathologies in RPE cells. The focus is on the process of light-induced oxidative degradation of pigment granules caused by reactive oxygen species. The reasons leading to increased oxidative stress in RPE cells as a result of the oxidative degradation of pigment granules are considered. A mechanism is proposed to explain the phenomenon of age-related decline in melanin content in RPE cells. The essence of the mechanism is that when the lipofuscin part of the melanolipofuscin granule is exposed to light, reactive oxygen species are formed, which destroy the melanin part. As more melanolipofuscin granules are formed with age and the development of degenerative diseases, the melanin in pigmented epithelial cells ultimately disappears.

## 1. Introduction

The retinal pigment epithelium (RPE) is a monolayer of epithelial cells closely adjacent on one side to the cells of the neural retina, and on the other side to the layer of choroidal capillaries [1,2,3]. On their apical surface, RPE cells have very long and thin microvilli, which project into the interphotoreceptor matrix, where they interact with the outer segments of photoreceptor cells, the rods and cones. On their basal surface, RPE cells are separated from the choriocapillaris layer by Bruch’s membrane (Figure 1). The RPE basal membrane has numerous folds and is part of Bruch’s membrane, which consists of five different layers and separates the RPE cells from blood vessels. This unique position of the RPE cells, which affects metabolite exchange between photoreceptor cells and blood vessels, determines the main functions of this tissue. These functions include the phagocytosis of shed photoreceptor outer segments, the transport and removal of metabolites from photoreceptor cells, the regulation of vitamin A metabolism and the control of the visual cycle, the absorption of scattered light, the regulation of ion flows, the production of growth factors for photoreceptors, providing ocular immune privilege by modulating the activity of immune cells (in particular microglial cells) within the retina and the maintenance of the blood–retina barrier [3,4,5].

Post-mitotic RPE cells undergo significant morphological changes with age. Thus, a decrease in the density of RPE cells is observed due to their loss [6,7].

**Figure 1 ijms-25-03609-f001:**
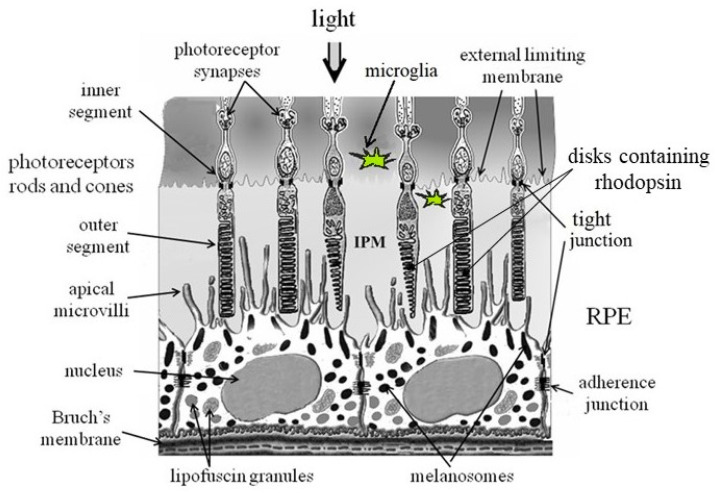
Scheme of the structure of the fundus. RPE—retinal pigment epithelium; IPM—interphotoreceptor matrix [8].

Outside the macular region, RPE cells increase in width and decrease in height, resulting in an overall thinning of the peripheral cell layer [9,10,11]. The cell cytoplasm decreases in volume and vacuolates, and the cells then become pleomorphic in terms of size, volume and content of nuclei and pigment granules [9,12]. Some RPE cells become multinucleated with age, especially outside the macular region [13]. Changes in the cytoskeleton of RPE cells are also observed. They lose their hexagonal shape and become larger and more elongated with age [14]. In addition, age-related RPE cells show an increase in the number of residual bodies (non-recyclable residues resulting from a disfunction of the phagocytosis process), the accumulation of basal deposits on Bruch’s membrane, a thickening of Bruch’s membrane, atrophy of RPE cell microvilli and the formation of drusen between the basement membrane of the RPE and the inner collagen layer of Bruch’s membrane [4,15]. However, despite changes in RPE cell structure and density during normal aging, the cell monolayer remains intact [16].

Changes in RPE cells become more pronounced with increased oxidative stress and the development of chronic ocular diseases such as cataracts, glaucoma, age-related macular degeneration (AMD), Stargardt disease and diabetic retinopathy [11,17,18,19]. Thus, AMD is characterized by the chronic and progressive degeneration of RPE cells, photoreceptors and retinal neurons [20]. In this case, the characteristic histological changes are expressed to a much greater extent than in aging RPE cells [21,22,23]. Characteristic histological features in RPE cells at early and intermediate stages of AMD are the presence of drusen on the basal side of the RPE along Bruch’s membrane, the accumulation of basal deposits within Bruch’s membrane, and pigment abnormalities within the RPE cells themselves [16,24,25]. In addition, extracellular deposits between photoreceptors and the RPE, called reticular pseudodrusen, are observed [25]. The deposition of soft drusen with age in the macular region of the retina is thought to precede the development of AMD and lead to vision loss [4].

The intracellular pigment abnormalities in the RPE observed with aging or the development of chronic diseases associated with oxidative stress are caused by the accumulation of lipofuscin-containing pigment granules and the depletion of melanin-containing granules [26,27,28,29,30].

## 2. Cytotoxic Properties of Lipofuscin and Its Role in the Development of Oxidative Stress in RPE Cells

Lipofuscin granules accumulate in RPE cells during aging and typically remain there until the end of life, and can occupy up to 30% of the volume of the RPE cell cytoplasm [31,32]. The intensity of LG accumulation in RPE cells increases with the development of a number of ocular pathologies, such as Stargardt disease, Best’s macular dystrophy and retinitis pigmentosa [19,33,34,35,36,37,38,39,40]. AMD has also been suggested to be associated with progressive LG accumulation [41,42,43]. There may also be an increased accumulation of complex melanolipofuscin granules (MLG) in AMD [44,45]. LG in the RPE is a by-product of phagocytosed shed photoreceptor outer segments, which accumulates as a result of incomplete lysosomal digestion [46]. LGs are heterogeneous and consist of a mixture of lipids, minor amounts of proteins modified and oxidized by end products of glycation and lipid peroxidation, and bisretinoid fluorophores that absorb light in the blue region of the spectrum and determine the autofluorescent properties of the granules [41,47,48,49,50,51]. Structurally, LGs are yellow-orange granules surrounded by a lipid membrane and have an average diameter of about 1.0 micron [52,53,54,55].

It is well known that LGs produce reactive oxygen species (ROS), such as superoxide radicals and singlet-state oxygen, when exposed to visible light, especially in the blue-green region of the spectrum [56,57,58,59,60]. ROS generated by LGs under the influence of light stimulate the oxidation of lipids and proteins and can cause the development of oxidative stress in the RPE [60,61,62,63,64,65]. For example, the irradiation of a culture of RPE cells loaded with lipofuscin granules leads to an increase in oxidized lipids and proteins and consequent damage to intracellular structures [43,66,67]. Oxidative stress is a chronic cellular condition in which pro-oxidant factors, such as ROS, suppress the activity of cellular antioxidant defense systems and initiate damage to intracellular proteins and lipids, leading to cell dysfunction. The phototoxicity of LGs is due to the presence in them of bisretinoid fluorophores, one of which, N-retinyl-N- retinylidene ethanolamine (A2E), has been shown to be localized not only in the RPE, but also in lysosomes [68,69], and to a lesser extent in mitochondria, the Golgi apparatus and the cytoplasmic membrane [70]. A2E may accumulate in RPE cells during aging [71] and act as an auto-oxidant by increasing oxidative stress [72] through the photogeneration of ROS such as superoxide radicals [73,74,75] and singlet oxygen [73,76]. In addition, due to its chemical structure, A2E can act as an amphiphilic detergent that can destroy the membrane structures of intracellular RPE organelles and induce apoptosis [60,77,78,79].

In the presence of oxygen, the irradiation of LG leads to the oxidation of bisretinoids and the formation of toxic products, namely epoxides, peroxides, aldehydes and ketones [80,81,82,83,84]. The formation of aldehydes and ketones in human LG irradiated with visible light in the blue-green region of the spectrum has been demonstrated by TOF-SIMS mass spectrometry and femtosecond broadband CARS [85]. Among the carbonyl products formed during the oxidation of LG, lipid peroxidation products such as reactive aldehydes and dialdehydes have also been found: 4-hydroxy-nonenal (4-NHE) and malondialdehyde (MDA) [48,86]. Active carbonyls formed both during the photoinduced oxidation of LG and the oxidation of LG by superoxide radicals can either remain inside lipofuscin granules (hydrophobic oxidation products) [81,84,87] or can diffuse into the cell cytoplasm (hydrophilic and amphiphilic oxidation products) [84]. It has been shown that water-soluble products of photoinduced oxidation of LG and A2E can modify water-soluble proteins as well as proteins and lipids of the outer segments of photoreceptors with the formation of fluorescent Schiff bases [84,88]. These water-soluble carbonyls are extremely toxic [89] and are believed to be precursors for the formation of advanced glycation end-products (AGEs) [90,91]. AGEs are suggested to be the initiators of the development of age-related cellular dysfunction, since they cause the formation of covalent protein cross-links, leading to a decrease in protein mobility and solubility, a decrease in enzymatic activity, and a loss of receptor recognition function [92,93]. It is known that AGE products such as pentosidine, carboxymethyllysine and carboxyethyllysine accumulate with age in significant quantities in Bruch’s membrane [94]. Oxidatively damaged molecules such as carboxyethylpyrrole, MDA, 4-NHE and AGEs can accumulate in the macular area and are sources of oxidative stress [95,96,97,98]. Damage to proteins and lipids during oxidative stress can lead to the inhibition of the process of utilization of photoreceptor outer segments phagocytosed by the RPE and an increase in protein resistance to lysosomal proteinases [97,99,100].

Modified and damaged proteins that cannot be repaired by heat shock proteins are utilized in proteasomes and by autophagy. Autophagy is also used by RPE cells to recycle damaged mitochondria. Autophagy is a complex lysosomal process of clearance to eliminate large damaged molecular structures such as intracellular organelles (mitochondria, endoplasmic reticulum, peroxisomes), ubiquitinated macromolecules and pathogens [16,101,102]. However, when the activity of lysosomal enzymes decreases, for example under oxidative stress, the autophagy process goes into decline, resulting in the accumulation of endogenous pathogenicity and danger signals (DAMPs) in cells, which can activate the inflammasome [103]. Endogenous molecular patterns associated with danger typically represent damaged macromolecules that can accumulate with age as a result of increased production (oxidative stress) and/or insufficient clearance [104,105]. The NLRP3 inflammasome is an important link in the development of the inflammatory response to tissue damage, and during aging, it is an important factor in the development of so-called “inflammaging”, when the immune reactions necessary for tissue repair turn into a chronic non-adaptive form as damage accumulates [106].

The NLRP3 inflammasome, present in human RPE cells [107,108], consists of three components: the cytoplasmic NOD-like receptor (NLR), the adapter protein ASC and the cysteine proteinase caspase-1 in an inactive form. When DAMP interacts with the inflammasome receptor (Figure 2), caspase is activated and converts the pro-inflammatory cytokine interleukin-1β (IL-1β) into its active form [109,110].

The activation of the inflammatory process manifests in its early stages as low-grade parainflammation associated with the recruitment of microglial macrophages [111]. The complement cascade is also activated by RPE cells [112,113,114], which can express various components and complement factors (C3, C5, CFF, CFH) which, in turn, trigger the secretion of proinflammatory cytokines [115,116]. Resident mononuclear phagocytes, microglia, are the dominant type of immune cells found in the nerve fiber layer, as well as in the inner and outer plexiform layers of the retina. They are thought to remove cellular debris from the subretinal space to protect photoreceptors and RPE from injury and death [117]. Activated microglial cells are recruited into the subretinal space, which promotes the removal of toxic aggregates [118].

The development of the inflammatory process in the early stages is successfully regulated by RPE cells due to their release of anti-inflammatory factors, including complement inhibitors and anti-inflammatory cytokines such as IL-10 [119,120]. However, aging and the progression of pathologies such as AMD are accompanied by the increased activation of microglia and the prolongation of their presence in the subretinal space, which potentially contributes to the damage of photoreceptor and RPE cells and consequently results in increased inflammatory processes [16,121]. When large amounts of DAMPs accumulate, the inflammatory process intensifies, the blood–retina barrier breaks down, and myeloid cells from the peripheral circulation are recruited and infiltrate from the bloodstream into the RPE, including monocytes, tissue macrophages and dendritic cells [122], which cause an inflammatory form of programmed cell death, pyroptosis [123].

All this indicates the importance of the balance between oxidative and antioxidant factors in RPE cells, regulating the development of oxidative stress. Melanin-containing organelles, which perform the function of protecting against light-induced damage in the retina, play a great role in the regulation of the oxidative balance in RPE cells.

## 3. Protective Role of Melanin-Containing Organelles in RPE Cells

The protective effect of melanin in RPE cells is associated with, firstly, its screening of the photosensitive elements of the retina by absorbing excess light and dissipating its energy in the form of heat; secondly, with the binding of both endogenous (photo)toxic molecules formed in the retina and RPE, and exogenous xenobiotics into inactive complexes; and thirdly with antioxidant and antiradical activities [8,124,125,126]. Melanosomes contain the pigment melanin, which is black or dark brown. Melanosomes are important pigment organelles of RPE cells. They develop from pre-melanosomes in the early stages of ontogenesis. Pre-melanosomes, in turn, are formed from endosomes of the Golgi complex [127]. Spherical or elongated melanosomes are observed in the RPE. The former are found mainly in the apical part of the cell and in microvilli, while the latter are localized in the middle part of the cytoplasm. In the basal region of RPE cells, melanosomes are rare. The size of the human RPE melanosome is 2.3 ± 0.5 µm in length and 0.9 ± 0.1 µm in diameter [128]. Human RPE melanosomes, unlike uveal melanocytes, contain mainly eumelanin-type pigment [129,130]. Eumelanins are irregular polymers containing indole-5,6-quinone monomer units in an oxidized or reduced state, and they have a stable ESR signal with a high concentration of paramagnetic centers.

Melanins in the eyes of vertebrates and humans perform the function of protection against the damaging effects of scattered light. They absorb light in a wide band of visible and ultraviolet irradiation, and the degree of absorption monotonically increases with the decreasing wavelength of light [124]. However, there are no absorption maxima or minima in either the visible or ultraviolet regions of the eumelanin spectrum. Most of the light energy absorbed by melanin is quickly converted into heat through the mechanism of internal conversion. Light that passes through the layers of the neural retina is absorbed by melanosomes located in the apical region of the RPE cells. It is estimated that the RPE absorbs about 34–60% of incident light in the foveal region and about 21–40% in the equatorial region [131]. As a result, the risk of potentially dangerous photochemical reactions is significantly reduced.

An absence or deficiency of melanin, albinism, is known to significantly increase the risk of light damage to the retina, primarily its photoreceptor cells and RPE cells. Albinos are extremely sensitive to the damaging effects of light [132]. Albino eyes are highly sensitive not only to light, but also to ischemia and various pro-oxidants [133,134]. All these effects appear to be associated with a deficiency or absence of melanin, a decrease or complete absence of its protective effect, including the screening of photosensitive structures from excess light [8,131], antioxidant protection [127,135,136,137,138] and protection from the action of toxic molecules, including bisretinoids, by their binding into inactive complexes [139,140].

As previously noted, RPE cells undergo significant biochemical and morphologic changes during the aging process, including the accumulation of the “age pigment” lipofuscin and complex pigment granules, such as melanolipofuscin granules, while simultaneously decreasing the number of melanin-containing melanosomes. While melanosomes occupy about 8% of the RPE cell volume before the age of 20 years, this volume gradually decreases to 3.5% between the ages of 41–90 years [30,31,141]. At ages 90–101 years, melanosomes are almost completely replaced by mixed melanolipofuscin granules [142,143]. It has also been shown that melanin pigmentation in the periphery of the retina declines with age [4,9,12]. This indicates the processes of age-related biodegradation of melanosomes in RPE cells. A decrease in melanin concentration in RPE cells is an important factor leading to increased oxidative stress in the cell [29,144,145,146]. The development of AMD is also accompanied by a decrease in the melanin content of RPE cells, and the melanin content is inversely proportional to the degree of AMD development [147,148]. An increase in oxidative stress with a decrease in melanin concentration may also be due to the fact that melanin degradation produces products with pro-oxidant properties [145,149,150,151]. The degradation of the melanin polymer molecule leads to a decrease in its antiradical activity and the appearance of fluorescent decomposition products exhibiting toxic and phototoxic properties [138,152,153,154,155,156].

## 4. Mechanisms of Age-Related Decrease in Melanin Concentration

Based on the known physicochemical properties of melanin, it can be assumed that it is resistant to degradation caused by enzymatic reactions. It is most likely that melanin degradation can be caused by exposure to light quanta or chemical oxidants such as ROS. Indeed, the oxidative destruction of melanin in melanosomes has been shown to occur with age. Thus, it has been shown that in elderly and old people there is an increase in fluorescence and an increase in the absorption of oxygen by RPE melanosomes [29,157]. The degradation of melanin by ultraviolet irradiation and/or hydrogen peroxide is known to produce fluorescent decomposition products [158,159,160]. It has been shown that melanin irradiation with intense visible light and ultraviolet leads to pigment degradation [161,162]. However, to achieve this effect, high irradiation energy and long exposures are necessary. Melanin irradiation with low-intensity light does not lead to its degradation [162]. It should be noted that the structures of the eye—the cornea, lens and vitreous body—transmit virtually no short-wave ultraviolet radiation to RPE cells containing melanin. It is also unlikely that visible light of such high intensity and duration, which is used to destroy melanin in the experiment, would ever affect the retina in vivo. Therefore, the degradation of melanin in RPE melanosomes for these reasons is usually excluded for the eye, but may occur for melanin in hair and skin exposed to direct sunlight.

On the other hand, oxidizing agents such as superoxide and hydrogen peroxide can cause melanin degradation relatively easily, and this is accompanied by a loss of antioxidant activity [145,155,162]. Thus, it has been shown that superoxide radicals cause the gradual destruction of melanosomes, accompanied by a drop in the ESR signal. When melanosomes are completely destroyed, a transparent solution is formed, paramagnetic centers in melanosomes disappear, and they lose their antioxidant properties [145]. As has been repeatedly mentioned, superoxide and other reactive oxygen species in large amounts can be formed in RPE cells by lipofuscin granules and bisretinoids under the influence of light, and are apparently the key molecules initiating the entire process of melanin degradation. It is highly likely that such a process may occur in mixed granules containing both lipofuscin and melanin simultaneously. Such complex granules, melanolipofuscin granules, contain on average 45% less melanin than melanosomes, as shown by ESR studies for pigment granules obtained from two age groups (Figure 3, Table 1) [145,163].

Since it is generally believed that MLG is formed by the fusion of a melanosome with a lipofuscin granule [31,142], this drop in melanin concentration in the MLG granule can be explained by the degradation of melanin [145]. Indeed, we have recently shown by mass spectrometry (ToF-SIMS), using principal component analysis (PCA), that the products of the oxidative degradation of RPE melanosomes caused by superoxide radicals in the dark are also present in the water-soluble fraction of blue-irradiated RPE melanolipofuscin granules [156]. It is logical to assume that this decrease is associated with the degradation of melanin in the MLG through interactions with ROS generated by the lipofuscin part of the granule (Figure 4, (1)).

A similar process of melanin degradation can occur in other melanin-containing granules if ROS generators are present in them simultaneously. Such generators can be either bisretinoids or photosensitive products of melanin degradation (PMD), which have recently been shown to be capable of the photo-induced generation of superoxide [156]. It is known that melanosomes in RPE cells can fuse with almost any phagocytosed material [166,167,168] including, as might be expected, bisretinoids formed during the visual cycle [169]. Due to the photooxidative properties of bisretinoids, they may also be involved in the degradation of melanin [168]. PMDs, which are formed during the partial degradation of melanin and are capable of the photoinduced generation of superoxide [156], can participate in the further degradation of the intact part of the melanin polymer and in the process of oxidation of lipofuscin (bisretinoids) (Figure 4, (2)).

It has recently been shown that when melanin is exposed to exogenous free radicals (superoxide, nitric oxide), it enters a high-energy state (“chemiexcitation”) in which the pigment causes the degradation of lipofuscin [164,165]. Such mechanisms for removing lipofuscin using activated melanin can lead to the degradation of the latter [165]. It can be speculated that the melanin polymer, contained in the granule simultaneously with bisretinoids and PMDs, changes into an excited state when exposed to light, which can lead to the autodestruction of the granule (Figure 4, (3)) and a decrease in the concentration of melanin in the RPE cell.

## 5. Conclusions

The role of melanosomes in the cells of the retinal pigment epithelium of the human eye cannot be overestimated. The disappearance of melanosomes (decrease in antioxidant protection) and the accumulation of lipofuscin granules (strengthening of ROS production) in RPE cells during aging and pathologies can lead to increased oxidative stress. Therefore, the excessive accumulation of lipofuscin granules in RPE cells, leading, among other things, to their accelerated fusion with melanosomes and the formation of mixed melanolipofuscin granules, is considered to be a significant pathogenetic factor. To explain the phenomenon of age-related decline in melanin content in RPE cells, we propose a mechanism in which, when light acts on the lipofuscin part of the melanolipofuscin granule, reactive oxygen species are formed which destroy its melanin part. Since more and more melanolipofuscin granules are formed with age and with the development of degenerative diseases, melanin in pigment epithelial cells ultimately disappears. The disappearance of melanin, as well as melanosomes themselves as screening light filters and antioxidants, significantly increases the risk of developing oxidative and especially photo-oxidative stress in the structures of the eye.

In future studies, it will be important to clarify the mechanisms responsible for the process of fusion of melanin and lipofuscin granules in RPE cells, and the role of Rab GTPases, motor proteins, light irradiation and reactive oxygen species in this process. In addition, it is important to investigate the roles of light irradiation, melanin and reactive oxygen species in the possible release of RPE lipofuscin granules into the extracellular space.

## Figures and Tables

**Figure 2 ijms-25-03609-f002:**
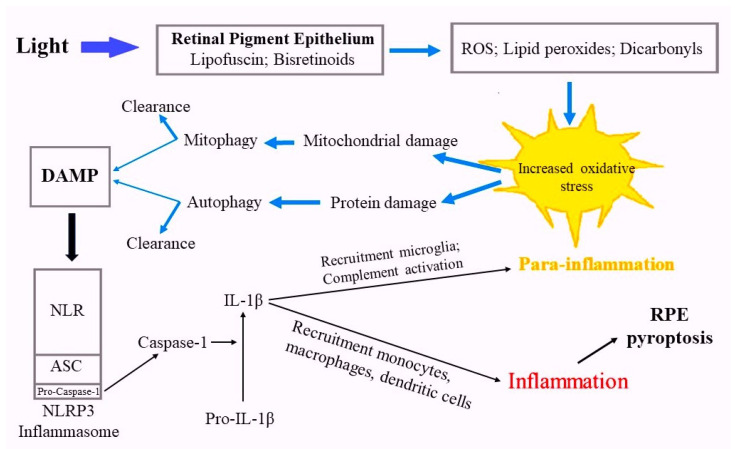
Schematic illustration of lipofuscin and bisretinoid’s roles in the development of photooxidative stress in the RPE cell.

**Figure 3 ijms-25-03609-f003:**
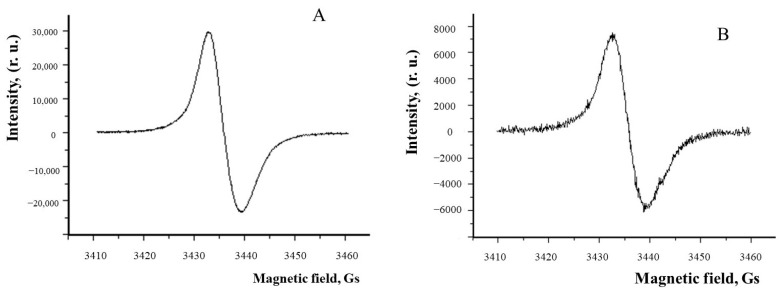
MLG from human RPE cells contain less melanin than MG. ESR spectra of MG (**A**) and MLG (**B**) from human RPE cells.

**Figure 4 ijms-25-03609-f004:**
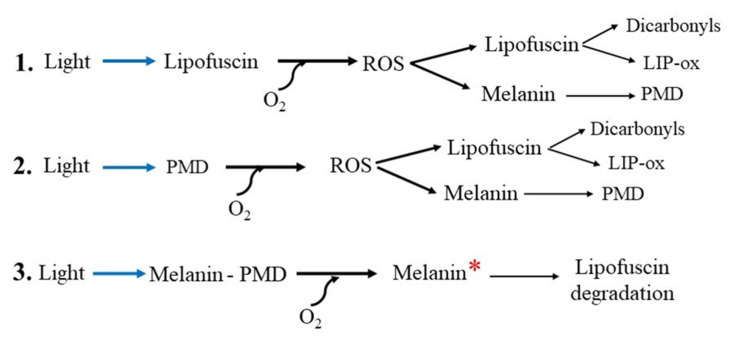
Scheme of the mechanisms involved in melanin degradation in the melanolipofuscin granule. Abbreviations: PMD—melanin degradation products, ROS—reactive oxygen species, LIP-ox—oxidized lipofuscin. (**1.**) Light in the presence of oxygen activates ROS generation mediated by lipofuscin fluorophores. The resulting ROS can oxidize both melanin, causing its degradation and the formation of PMD, and lipofuscin, causing the formation of reactive dicarbonyls. (**2.**) The resulting photosensitive melanin degradation products (PMDs) generate ROS when exposed to light and, in turn, cause the further degradation of melanin and lipofuscin. (**3.**) In a granule containing all three components, namely melanin, lipofuscin (bisretinoids) and PMD, light and ROS activate the transition of melanin to a high-energy state (melanin*) in which the excited pigment causes the degradation of lipofuscin [164,165].

**Table 1 ijms-25-03609-t001:** Comparison of melanin concentrations in MG and MLG from RPE cells for donors of two age categories.

Age Groups	Melanin Concentration, mg/per Granule	
	Melanosomes (MG)	Melanolipofuscin (MLG)
30–60 years old, (50 eyes)	(2.3 ± 0.4) × 10^−10^	(1.3 ± 0.5) × 10^−10^
60–75 years old (30 eyes)	(2.2 ± 0.8) × 10^−10^	(1.2 ± 0.4) × 10^−10^

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
