# Peer review of "Retinal Pigment Epithelium Pigment Granules: Norms, Age Relations and Pathology"

_ijms, 2024, doi:10.3390/ijms25073609_

Round 1
Reviewer 1 Report
Comments and Suggestions for Authors
The review analyzes the functional role of the pigmented granules of the retinal pigmented epithelium in aging and develops three main aspects:
1. The cytotoxic properties of lipofuscin and its role in the development of oxidative stress; 2. The protective role of melanin; and 3. The mechanism of the age-related decrease in the melanin concentration. All are related to the retinal pigment epithelium.
The authors consider that when the lipofuscin of the melanolipofucsin granules is exposed to light, the reactive oxygen species developed destroy melanin. With aging, the melanin in the epithelium finally disappears.
The topic covered is interesting for understanding the physiopathology of some ophthalmic diseases. The review is clear, relevant, and well-written.
Tables and figures are appropriate, easily interpretable, and aligned with the text. Conclusions are supported by the references.
The cited references are appropriate and relevant, although only 18% have been published within the last five years. The authors' main research subject is the pigmentary epithelium granules, which could explain the 12%
self-citations.
Reviewer 2 Report
Comments and Suggestions for Authors
This is a brief and useful review fosuing on the physiology and possible mulfunctioning of melaning granules (and lipofuscin deposits) in the RPE, with an implicit reference to mammalian and human organization of this tissue. A mechanism is proposed to explain the deterioration of melanin granules with aging, associated to increased oxidative stress in the RPE.
The manuscript is highly concentrated on granules function and overlooks the (many and complex) functions of the RPE in retinal homeostasis; however, the brief review is self contained. Some indications are provided to facilitate reading and to broaden the horizon of the topic treated:
1) Please add histological illustrations showing melanin granules, lipofuscin deposits, general organization of the RPE, using biological images rather than diagrams.
2) When explaining the many roles of the RPE, please mention clealry the fact that tthis layer regulates the behavior of immune cells within the retina, which can lead microglial cells to reduce inflammation and promote immunological tolerance. This properties changes with aging and might be strictly related to other alterations you mention in the granules compartment.
3) If possible, write a brief section in which you explain how the RPE is studied for research purposes (i.e. measuring TEER, using ARPE 19 cells, using immunofluorescence etc.) and, if of use, in clinical approaches.
4) state where future research is going, or what is striving at, and what we are expected to learn in the near future by novel approaches to the RPE.
Comments on the Quality of English Language
TThe quality of the language is good and the text clear.
